# Pre- versus Post-Menopausal Onset of Overactive Bladder and the Response to Vaginal Estrogen Therapy: A Prospective Study

**DOI:** 10.3390/medicina59020245

**Published:** 2023-01-27

**Authors:** Yoav Baruch, Marco Torella, Sarah De Bastiani, Michele Meschia, Massimo Candiani, Nicola Colacurci, Stefano Salvatore

**Affiliations:** 1Urogynecology Unit, Tel Aviv Sourasky Medical Center, Tel Aviv University, Tel Aviv 6997801, Israel; 2Department of Obstetrics and Gynecology, University of Campania “Luigi Vanvitelli”, 80128 Naples, Italy; 3Obstetrics and Gynecology Unit, Vita-Salute University and IRCCS San Raffaele Hospital, 20132 Milan, Italy; 4Urogynecology Service, Centro Medico Boccaccio, 20123 Milan, Italy

**Keywords:** menopause, overactive bladder, GSM, estrogen therapy, GPFSBQ, PPIUS, VHI

## Abstract

*Background and Objectives:* This study examined the utility of local estrogen therapy for improving urinary symptoms in women diagnosed with Overactive Bladder allied to the time of onset of urinary symptoms whether pre- or post-menopausal. *Materials and Methods*: Subject to informed consent, menopausal women diagnosed with Overactive Bladder (OAB) and Genitourinary Syndrome of Menopause (GSM) were enrolled at three urogynecological units. OAB symptoms were scored using the Global Pelvic Floor Symptoms Bother Questionnaire (GPFSBQ), with explicit attention to question number 3 that specifically addresses the presence or absence of urgency and the Patient Perception of Intensity of Urgency Scale (PPIUS). The Vaginal Health Index (VHI) was used to assess the vaginal mucosa trophism. Exclusion criteria included: Pelvic organ prolapse (POP) ≥ stage II, urinary tract infection or disease, diabetes, inflammatory diseases, use of diuretics, alcohol or drug addictions, neurological and/or psychiatric disorders, and other precluding conditions. Women were treated with local estrogens for 3 months and re-evaluated. *Results*: Forty-three post-menopausal women were enrolled. Of these, ten women developed OAB symptoms before menopause (Group I) and 33 developed symptoms after menopause (Group II). Following local estrogen therapy, based on the Global Pelvic Floor Symptoms Bother Questionnaire, improvement of OAB symptoms was reported by 20% of patients in Group I (*p* = 0.414) and 64% of patients in Group II, (*p* = 0.002). Based on the PPIUS scale, diminution in urinary urgency was experienced by 20% of patients in Group I (*p* = 0.68) and 66% of patients in Group II (*p* = 0.036). Improved VHI scores were graded statisticaly significant in both groups (Group I in 100% of women, *p* = 0.005 vs. 76% in Group II, *p* = 0.004). *Conclusions*: Our results indicate that local estrogen therapy is more effective in women who develop OAB after menopause.

## 1. Introduction

After menopause, women commonly experience a variety of symptoms attributed to hormonal deprivation and primed by physiological and psychological age-related changes. Estrogen deficiency causes vulvovaginal atrophy heralded by symptoms such as dyspareunia, irritation, vulvar itching, bleeding, vaginal dryness, and burning sensation. These symptoms may be accompanied by urinary symptoms including frequency, urgency, nocturia, and incontinence [1]. Vulvovaginal atrophy is frequently called atrophic vaginitis, urogenital atrophy, or Genitourinary Syndrome of Menopause. The term Genitourinary Syndrome of Menopause (GSM), introduced in 2014, encompass genital (dryness, itching, and burning in the vagina), urinary (incontinence and recurrent urinary tract infections), and sexual symptoms (low sexual desire, poor lubrication, and dyspareunia) that may disturb quality of life [2]. Urinary incontinence, yet another common condition, incorporates two main types, stress and urge urinary incontinence [3]. Overactive Bladder (OAB) syndrome alludes to the presence of urgency, with or without urinary incontinence, commonly associated with nocturia and increased occurrence of urination and urge incontinence in the absence of urinary tract infection or other diseases [3,4,5]. OAB is highly prevalent in the population at large, ranging from 10% to 45% dependent on the population studied [3]. OAB is linked with higher propensity for adverse health conditions and diminution in quality of life. The prevalence of OAB rises with age which by itself is considered a major risk factor for the development of the condition. OAB’s etiology is multifactorial and linked to issues influenced by anatomic changes, lifestyle-related factors, comorbidities, and genetic susceptibility. Treating this highly prevalent condition is challenging, especially in the older population. As urinary tract infection and OAB share common features such as frequency, urgency, and nocturia women presenting with OAB symptoms are often diagnosed and treated as urinary tract infections without performing a urine culture. When urine cultures are acquired, they may result positive in part of these women indicating that empiric therapy of urinary tract infections frequently lead to a missed diagnosis [3]. In such cases, detailed anamnesis and urinalysis scan can secure and establish the correct diagnosis.

A stepwise wide approach to the treatment of OAB symptoms include changes in lifestyle, pelvic floor muscle and bladder training, drug therapy, and neuromodulation. First-line treatment of OAB involves behavioral therapy aimed to reduce urinary frequency and urgency and to increase bladder outlet volume capacity. This intention can be reached by managing fluid intake and instructing women with OAB to engage in pelvic floor muscle and bladder training. These consist of repeated voluntary contractions of the pelvic muscles. Strengthening pelvic and bladder musculature is imperative in order to elevate the pelvic floor and the position of the bladder. Other measures include avoidance of bladder irritants, treatment of constipation and formulation of weight loss trials [6,7,8]. Second-line therapies for OAB include pharmacological treatment with muscarinic receptor antagonists, beta adrenergic receptor agonists or their combination which are delivered orally in most cases. These drugs are effective in controlling urgency, urge incontinence, and nocturia. Third-line treatment include intra-detrusor onabotulinumtoxin A and sacral neuromodulation in carfully selected patients. Recently, vaginal laser therapy was proposed as a minimally invasive effective therapy for women with OAB. Individualized fourth line treatment options are rarely used.

Seeking to avoid troublesome side effects alternative delivery routes other than oral (transdermal, intra-vesicular, vaginal, and intramuscular) have been employed [9]. The vaginal route proved efficiant in maintaining extended drug release and improving patients’ compliance [9]. Bioadhesive vaginal gels are safely used to simultaneously treat vaginal dryness and overactive bladder after menopause [9].

The lower genital and lower urinary tract share similar embryologic origin and contain abundant estrogen receptors [3]. Hormonal replacement therapy is beneficial and effective in the management of menopausal urinary symptoms and urogenital atrophy [10,11,12,13]. However, some women are reluctant to use oral estrogens as these have been linked to an increase in the risk for breast cancer and thrombotic events. Otherwise, intravaginal estradiol tablets seem to be operative for the treatment of lower urinary tract symptoms and for the attenuation of sensory urgency, frequency, stress, and urge urinary incontinence [10,11,12,13].

The effectiveness of estrogen in treating urinary incontinence and OAB symptoms was previously examined and considered [11]. Yet, information on the usefulness of local estrogen therapy, based on the age of onset, whether pre- or post-menopausal, is currently not available.

This study sought to compare the effects of local estrogen therapy on vulvo vaginal atrophy and urinary symptoms in women with GSM and OAB who were stratified by menopause status (pre-menopausal or post-menopausal) at the time of OAB onset.

## 2. Materials and Methods

The study was approved by the participating Institutional Review Boards and eligible women with OAB were recruited at the urogynecological outpatient clinics of three participating centers, namely: IRCCS San Raffaele Hospital (Milan), Fornaroli Hospital (Magenta), and Santa Maria degli Incurabili Hospital (Naples). All the participants signed written informed consent forms.

Inclusion criteria: menopause interval greater than 12 months, symptoms relatable to OAB (urgency, frequency, nocturia with/without incontinence) with or without symptoms pertinent to vulvovaginal atrophy. Women had to be ambulatory, community dwelling, and compliant with follow up. OAB is a clinical diagnosis and as such urodynamic testing was not deemed mandatory. Only patients who were not previously treated with estrogen or other alternative treatments for OAB were recruited. Exclusion criteria include: pelvic prolapse ≥ II grade according to Pelvic Organ Prolapse (POP) System (POP-Q) classification, current UTI, history of cancer, pelvic irradiation, neurological condition affecting bladder function, diabetes, chronic inflammatory diseases, chronic kidney desease, hypertension, congestive heart failure, use of diuretics, alcohol or drug addictions, and uncontrolled psychiatric disorders. POP ≥ II grade may cause, other than bladder outlet obstruction, lower urinary tract symptoms, and bladder complaints mimiking OAB symptoms [14] and as such women with POP ≥ II grade were excluded from the study.

Medical history and anamnesis were obtained. All participants underwent a physical examination and vulvovaginal atrophy was graded using the Vaginal Health Index (VHI). VHI scores five components namely: vaginal pH, elasticity, fluid secretion, epithelial integrity, and moisture on a scale of 1 (worst) to 5 (best) [15]. Scores ≤ 15 denote vaginal atrophy. Lower scores denote greater atrophy. Concomitantly, patients completed the Patient Perception of Intensity of Urgency Scale (PPIUS) and the Global Pelvic Floor Symptoms Bother Questionnaires (GPFSBQs).

PPIUS is a reliable and valid patient-reported outcome tool employed to measure urinary urgency and urge incontinence in OAB syndrome [16,17,18]. The level of urgency was graded by a 5-point scale. Responses range from “No urgency” to “Mild urgency,” “Moderate urgency,” “Severe urgency,” and “Urge incontinence.” Patients with 0 level of urgency at baseline were excluded from the study. GPFSBQ is a valid and reliable a self-administered assessment tool set to grade pelvic floor symptoms and the level of bother from various pelvic floor symptoms [19]. In a recent systematic review, the reliability of GPFBQ was validated and graded “sufficient” for good measurement of properties (intraclass correlation coefficient greater than 0.7) [20].

Attention was given to question number 3, referring to the presence or absence of urgency: Do you experience an abnormal strong feeling of urgency to urinate (sudden, compelling urge to void). Proposed answers: Yes or No and if yes, how much does it bother you on a 5-point scale? 1 = Not at all, 2 = Only a little bit, 3 = Somewhat, 4 = A moderate amount, 5 = A lot. Women who answered “Only a little bit” and greater bother scores were considered to embody OAB symptoms and were included in the study. Patients with no level of bother were excluded.

Participants were prescribed vaginal estrogens 50 mcg estriol in 1 g of vaginal gel (Italfarmaco S.A) one application per day for 21 consecutive days and then twice a week for another 9 weeks. After 12 weeks of treatment patients were re-evaluated at the outpatient clinic by the same examiner. Vulvovaginal atrophy was graded using VHI and participants were asked to complete the PPIUS and the GPFSB questionnaires.

The aim of this study was to compare the response to local estrogen therapy, in women with vulvovaginal atrophy and OAB categorized by menopause status (pre- or post-menopausal) at the onset of OAB symptoms. Accordingly, the cohort was stratifyed into two groups. Group I included participants who developed OAB before menopause. Group II included participants who developed OAB after menopause. Scores were calculated at baseline and after the completion of treatment. The difference between scores (delta) were compared between Groups I and II.

Considering a mean of prevalence of 35% for OAB (between 27% and 43%) and a width of final confidence interval (CI) of 0.15, it was calculated that at least 15 patients are needed for a confidence interval of 95% [21,22,23,24]. A significant sample requires a ratio of 1:3 women with pre- and post-menopausal onset of symptoms [25].

Statistical Analysis: Categorical variables were reported as frequency and percentage. Continuous variables and ordinal variables were reported as mean and standard deviation (SD) for reader ease. Continuous variables and ordinal variables were compared between pre- and post-menopausal women using Mann–Whitney tests. Continuous variables and ordinal variables were compared pre- and post-treatment using Wilcoxon test. All statistical tests were two sided and *p* value of less than 0.05 was considered as statistically significant. SPSS software (IBM SPSS statistics, version 28, IBM Corporation, Armonk, NY, USA, 2021) was used for all statistical analyses.

## 3. Results

A total of 43 patients affected by OAB syndrome were enrolled. Of these, 10 women (mean age 65.4 ± 8; range 54 to 78 years; Group I) endured symptoms of OAB before menopause and 33 experienced symptoms of OAB after menopause (mean age 62.4 ± 7; range 52 to 74 years; Group II). There were no statically significant age-related differences between the groups (*p* = 0.35).

### 3.1. VHI Scores

After 12 weeks, a statistically significant improvement was noted for all groups. In Group I the baseline VHI scores increased after therapy in 100% of patients (10/10) and in Group II baseline VHI scores increased after therapy in 76% of patients (25/33). Differences between groups are noted in Table 1.

### 3.2. GPFSBQ

As shown in Figure 1 and presented in Table 1: After 12 weeks, amelioration in symptom bother score of participants from Group II (post-menopausal onset of OAB) was significantly greater than the improvement attained by participants from Group I (premenopausal onset of OAB). Scores obtained in response to question number 3 (grading urgency) remained unchanged for Group I at 12 weeks (within the range of 4 “a moderate amount” and 2 “only a bit”) whereas in Group II scores decreased (from within the range of 5 (“a lot”) and 1 (“not at all”) to a range of 5 (A lot) and 0 (absence of urgency). Reduction in bother scores was gained by 21 women in Group II (21/33; 64%) and only by two women (2/10; 20%) from Group I. Differences resulted statistically significant.

### 3.3. PPIUS

As shown in Figure 2 and presented in Table 1: the scores obtained by participants from Group I (premenopausal onset of OAB) ranged from 1 (“mild urgency”) to 4 (“urgency incontinence”) before treatment and remained unchanged after treatment. The scores obtained by participants from Group II (post-menopausal onset of OAB) ranged from 1 (“mild urgency”) to 4 (“urgency incontinence”) before treatment and from 0 (“no urgency) to 4 after treatment. After treatment, only two patients from Group I had lower scores (2/10; 20%) compared with 22 patients from Group II (22/33; 66%). The decrease in urgency score after 12 weeks was noted only in Group II.

## 4. Discussion

OAB is a complex multifactorial condition that affects as much as 15% of women in general and as much as 43% of women aged 65 years and older [21,22,25,26,27]. OAB may result from neurogenic dysfunction or other age-related alterations in bladder role. It is acknowledged that one in two women suffers from vulvovaginal atrophy related to menopause.

The pathophysiology of OAB is not clearly defined. OAB can be related to either myogenic or neurogenic causality. Otherwise, post-menopausal estrogen deprivation seems to play a prominent role in the causation of OAB. It has been clearly shown that women who have gone through menopause frequently have lower a capacity bladder, impaired detrusor contractility, and reduced urine flow rates [10]. Estrogen deprivation per se is responsible for atrophy of the urogenital tract and is linked to incontinence, urinary frequency, nocturia, urgency, and recurrent UTIs. Age-related OAB in the elderly population can be managed safely and effectively with estrogen that thereby contributes to enhance patients’ quality of life. Estrogen is known to directly affect detrusor function and its deficiency might trigger the inception of urinary symptoms. Up to 70% of women link the onset of lower urinary symptoms with the end of their menstruation, suggesting that estrogen insufficiency plausibly cause the onset of symptoms [28].

Estrogen therapy is offered to women who experience symptoms of OAB. Estrogen improves the elasticity and the strength of the muscles in the pelvic floor thereby expanding bladder control and raises the bladder’s sensory threshold thereby reducing urinary incontinence. Altogether, estrogen can help to reduce the frequency of urination and to manage OAB.

In menopausal OAB patients, local estrogen therapy was demonstrated to significantly reduce symptom distress and is highly effective in alleviating GSM symptoms. Vaginal estrogen, operative for the treatment vulvovaginal atrophy, may provide relief to post-menopausal women with OAB [9,10]. Estradiol has been shown to reverse bladder atrophy [29]. Even though the role of local estrogen therapy in the management of vulvovaginal atrophy has been established, it remains controversial in the management of OAB, a condition frequently misdiagnosed with recurrent UTIs.

In women with lower urinary tract symptoms, local estrogen therapy was shown to diminish urgency, frequency, nocturia, and enhance bladder capacity and first sensation to void [10,11,30]. Estrogen therapy conceivably eases symptoms of OAB by raising bladder’s sensory threshold, enhancing urethral smooth muscle’s sensitivity to alpha-adrenergic receptors, encouraging beta 3 adrenoceptor-mediated detrusor muscle relaxation and by boosting collagen’s quality and construction in the peri-vesical and peri-urethral areas [1,10].

Both genetic and environmental effects account for the susceptibility to urinary incontinence, frequency, and nocturia in women. Shared environment however seems to be more influential for the predisposition to develop OAB. A systematic review completed to detect publications reporting gene expression in human bladder tissue specimens identified 11 genes as being up- or downregulated among patients with OAB vs. healthy individuals. These are included amongst other genes involved in synaptic transmission and smooth muscle contraction [31].

The response to vaginal local estrogen therapy in women with vulvovaginal atrophy and OAB stratified by menopause status at the onset of OAB symptoms was studied and reported here for the first time. Improvement of OAB symptomatology proved to be greater in women who endured overactive bladder symptoms after they entered menopause than in those who developed OAB before menopause.

The purpose of this study was to assess the effect of vaginal estrogen on the amelioration of OAB symptoms related to their onset, either before or after menopause. In order to investigate close to or net pre- and post-menopausal OAB women, we excluded all women with comorbidities that can contribute to urinary symptoms. Following therapy with vaginal estrogens OAB subjective symptom severity scores improved with significant differences noted between Groups I and II in favor of women who developed OAB symptoms after menopause. Local estrogen therapy was linked to enhancement of VHI scores in both Group I and II with no statistical difference between groups. Subjective symptom impression is an important primary outcome and is linked to patients’ perception of efficacy and their long-term compliance to treatment. The results of this study show attenuation of bother and severity of urge in the post-menopausal group and as such the effectiveness of estradiol therapy might be highlighted in women who develop OAB after menopause. This preliminary observation needs to be expanded and confirmed. This study meets sample size requirements and indicates that measured changes are meaningful. It is limited by its sample size that was rather restricted by the fact that it was difficult to recruit women with OAB who were not previously treated.

Guidelines for the management of OAB do not currently include low dose vaginal estrogen [32]. Further study is necessary to better understand the effect of local estrogen therapy and stipulate the ideal time for its use.

## 5. Conclusions

The results of this prospective study, although limited in magnitude, argue in favor of administrating vaginal estrogen to treat OAB. Based on our results, local estrogen therapy seems to be more effective in treating women who develop OAB after menopause than in women who endure OAB before menopause. The concept that OAB presenting after menopause, rather than before menopause, is correlated to estrogen deficiency needs to be affirmed, confirmed, and supported by future studies.

## Figures and Tables

**Figure 1 medicina-59-00245-f001:**
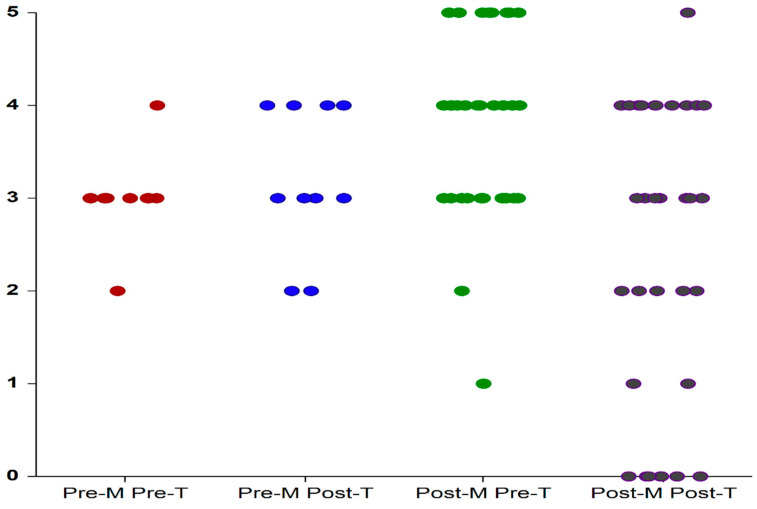
GPFSB3 score before treatment (Pre-T) and after treatment (Post-T) with local estrogens in patients with OAB of premenopausal (Pre-M) and post-menopausal (Post-M) onset. Each dot depicts an individual patient. T: treatment; M: menopause.

**Figure 2 medicina-59-00245-f002:**
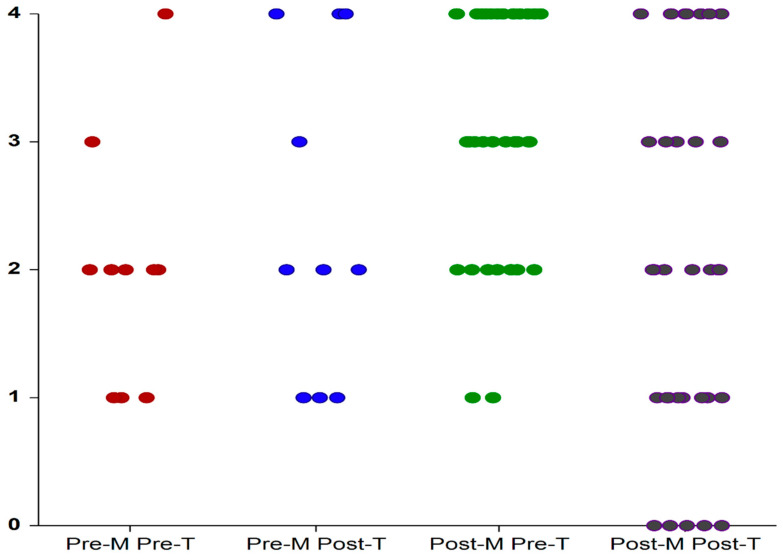
PPIUS values before treatment (Pre-T) and after treatment (Post-T) with local estrogens in patients with OAB symptoms of premenopausal (Pre-M) and post-menopausal (Post-M) onset. Each dot depicts an individual patient. T: treatment; M: menopause.

**Table 1 medicina-59-00245-t001:** Comparison between pre- and post-menopausal onset of overactive bladder.

	Pre-Menopausal	Post-Menopausal	*p*
Age	65.4 ± 8	62.45 ± 7.1	0.356
VHI			
Pre-treatment	11.6 ± 2	11.6 ± 3.1	0.989
Post-treatment	17.8 ± 1.2	14.12 ± 3.5	<0.001
delta VHI	6.2 ± 2.6	2.48 ± 3.5	0.009
*p* _within groups_	0.005	0.004	
GPFSB-3			
Pre-treatment	3.0 ± 0.47	3.79 ± 1	0.013
Post-treatment	3.2 ± 0.79	2.48 ± 1.6	0.341
delta GPFSB-3	0.2 ± 0.79	−1.3 ± 2	0.026
*p* _within groups_	0.414	0.002	
PPIUS			
Pre-treatment	2.1 ± 0.87	3.06 ± 1.03	0.015
Post-treatment	2.3 ± 1.33	2.33 ± 1.47	0.944
delta PPIUS	0.2 ± 1.31	−0.7 ± 1.77	0.184
*p* _within groups_	0.680	0.036	

NS: statistically significant, *p* < 0.05.

## Data Availability

Data supporting reported results can be found in a specific archived database generated during the study and available upon reasonable request.

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
