# Peer review of "Pre- versus Post-Menopausal Onset of Overactive Bladder and the Response to Vaginal Estrogen Therapy: A Prospective Study"

_medicina, 2023, doi:10.3390/medicina59020245_

Round 1

Reviewer 1 Report

This research has some merits and proposes the advantage of intravaginal estrogen therapy for ameliorating OAB in postmenopausal women. However, this manuscript needs further amendment. Reviewer suggests resubmission after careful revision.

Authors used the terms of topical approach (line 65), local estrogen therapy (line 71), and vaginal estrogen therapy (line 74). These may be confusing for readership. Authors should either use consistent term or describe each term specifically.

Reliability analysis of this study should be conducted and be stated in section of result.

Ethical statement including approval from the local institutional review board (IRB) or other appropriate ethics committee must be described in the section of method.

The characteristics of different types of intravaginal therapies for OAB could be discussed. Authors may refer to these references. (DOI: 10.1016/j.mayocp.2019.11.024) (doi: 10.3390/ph14050409)

Line 128. Define “CI”.

Please state statistical analysis software.

In the section of results, the basic characteristics of patients must be stated in table.

Line 142, 143. “There were no statically significant differences between the age groups (p=0.35)” Line 149 “The difference between groups was not statistically significant (p = 0.20).”

Please illustrate the data in table, and please state the analysis methods and all of the values.

Figure legend should be placed beneath the figure. Please check submission guideline.

Figure 1 (result 3.2. GPFSBQ) and figure 2 (result 3.3. PPIUS) should be illustrated in column chart with average score, error bar, and significant symbol. One-way ANOVA should be used, rather than chi-square test.

Line 235. The conclusion is too brief. The regular rule should be about 100-150 words.

A few grammatical mistakes or writing errors can be revised.

Line13. utility of local estrogen therapy “for”

Define OAB, GSM, POP in abstract.

Line 22. “reevaluated”

Line 24. “and in 33” needs grammar check.

Line 33. “symptoms that were attributed to”

Line 41. Just “stress and urge urinary incontinence”

Line 43. “incontinence, which is usually associated…”

Line 44. “and nocturia, and may…”

Line 47. “History and urinalysis scan establish the diagnosis” should be more specific.

Line 48. “out as OAB because UTI share the common features” is more appropriate.

Line 66 “Although, in general, …” , line 67 “been imperial in…” and line 70 “Although, several studies…” are kind of odd.

Line 67. “, whereas intravaginal estradiol tablets seem to be more operative…” would be better.

Line 81. “Participants provided written informed consent forms” Please check grammar.

Line 97. “Lower is the score” ?

Line 105. “reliable a self-administered 9-item” Please check again.

Line 116. “After 12 weeks of treatment, patients…”

Line 144. “After 12 weeks, a reduction…”

Line 146. “After 12 weeks, greater improvement…”

Author Response

Reviewer 1: Response to Comments and Suggestions

Comment no. 1: This research has some merits and proposes the advantage of intravaginal estrogen therapy for ameliorating OAB in postmenopausal women. However, this manuscript needs further amendment. Reviewer suggests resubmission after careful revision.

Response to comment no. 1: We hereby resubmit the MS after carefull amendment taking into consideration the suggestions made by the distinguished reviewer.

Comment no. 2: Authors used the terms of topical approach (line 65), local estrogen therapy (line 71), and vaginal estrogen therapy (line 74). These may be confusing for readership. Authors should either use consistent term or describe each term specifically.                                           Response to comment no. 2: We find best appropriate the the term "local estrogen therapy" and this term is now used consistenthly throughout the MS. 

Comment no. 3: Reliability analysis of this study should be conducted and be stated in section of result.

Response to comment no. 3: In this study we used two well validated questionnaires  and as such we hope that reliability analysis is not further required. References are alluded to in the text.

Comment no. 4: Ethical statement including approval from the local institutional review board (IRB) or other appropriate ethics committee must be described in the section of method.

Response to comment no. 4: IRB approval statement is included in the method's section.

Comment no. 5: The characteristics of different types of intravaginal therapies for OAB could be discussed. Authors may refer to these references. (DOI: 10.1016/j.mayocp.2019.11.024)(doi:10.3390/ph14050409).                                                                                                                  Response to comment no. 5: The references were assimilated  as requested (references 6 & 9 in revised MS).

Comment no. 6: Line 128. Define “CI”.

Response to comment no. 6: CI was defined.

Comment no. 7: Please state statistical analysis software.

Response to comment no. 7: As requested statistical analysis is now specified in the method's section. 

Comment no. 8: In the section of results, the basic characteristics of patients must be stated in table.                                                                                                                                                 Response to comment no. 8: age related data provided as requested.  

Comment no. 9: Line 142, 143. “There were no statically significant differences between the age groups (p=0.35)” Line 149 “The difference between groups was not statistically significant (p = 0.20).”                                                                                                                                            Response to comment no. 9: recalculated p values are presented in table 1 as requested.

Comment no. 10: Please illustrate the data in table and please state the analysis methods and all the values.                                                                                                                                       Response to comment no. 9:  Provided as requested.

Comment no. 10: Figure legend should be placed beneath the figure. Please check submission guideline.                                                                                                                                         Response to comment no. 10:  Figure legends are now placed beneath the figures.

Comment no. 11: Figure 1 (result 3.2. GPFSBQ) and figure 2 (result 3.3. PPIUS) should be illustrated in column chart with average score, error bar, and significant symbol. One-way ANOVA should be used, rather than chi-square test.                                                                   Response to comment no. 11: Figure 1 depicts crude values rather than summarized data. The average scores (and SD) are presented in tables, as requested, so the reader can benefit from both summarized and crude data. The statistical analyses were re-performed using different tests as described in the current version of the method's section. 

Comment no. 12: Line 235. The conclusion is too brief. The regular rule should be about 100-150 words.                                                                                                                                                Response to comment no. 12: The conclusion section is now more comprehensive

Comment no. 13: A few grammatical mistakes or writing errors can be revised.                            Response to comment no. 13: We carefully amended all mistakes and writing errors as kindly suggested (Lines 13, 22, 24, 33, 41, 43, 44, 47, 48, 66, 67, 81, 97, 105, 116, 144, 146 in original version).

Comment no. 14: Define OAB, GSM, POP in abstract.                                                                              Response to comment no. 14: We Defined OAB, GSM, POP in abstract as requested

Specific remarks: Lines 116, 114, 116. “After 12 weeks of treatment, patients…”.                      Response: The treatment interval was indeed 12 weeks-Participants were prescribed vaginal estrogens 50 mcg estriol/1gram vaginal gel (Italfarmaco S.A) one application per day for 21 days then twice a week for another 9 weeks altogether 12 weeks..

Reviewer 2 Report

Introduction:

I believe the introduction section of the paper could be improved and made clearer to better convey the information to the reader. As it is currently written, I find the introduction to be somewhat confusing and difficult to understand. It would be helpful if the authors rewrote the introduction in a more organized and coherent manner, explaining the main points and ideas they are trying to convey to the reader in a clear and concise way. This would make the paper more accessible and engaging for readers and help them to better understand the information being presented.

The effectiveness of estrogen in treating urinary incontinence and overactive bladder symptoms has been studied in several clinical trials (for example: https://clinicaltrials.gov/ct2/show/NCT02524769). These studies have generally found that estrogen therapy can be effective in improving symptoms of UI and OAB in some women. Estrogen works by improving the elasticity and strength of the muscles in the pelvic floor, which can help improve bladder control and reduce the frequency of urinary incontinence. In terms of the effectiveness of local estrogen therapy, based on the age of onset of urinary symptoms, this may vary depending on the individual and the specific symptoms they are experiencing. Some studies have found that local estrogen therapy is more effective in treating symptoms of UI and OAB in women who are experiencing these symptoms at a younger age. However, the best approach to treatment will depend on the individual and their specific symptoms and health needs.

1.      Menopause senescence is a term that is used to describe the natural process of aging that occurs in women as they approach and go through menopause. This process is characterized by a decline in the levels of the hormone estrogen, which can lead to a number of physical and psychological changes in the body. It is a natural part of the aging process and is not considered a medical condition. Please elaborate that in the introduction.

2.      The term genitourinary syndrome of menopause (GSM) refers to a group of symptoms that can affect the genitals and urinary system in women who are going through or have gone through menopause. These symptoms can include dryness, itching, and burning in the vagina, as well as urinary incontinence and recurrent urinary tract infections. These all should be highlighted in the introduction section of the paper.

3.      I find the explanation of UI, GSM, and VVA in the introduction section of the paper to be unclear and lacking coherence, which may make it difficult for readers to pay attention and understand the information being presented. It would be helpful if the terms were defined and explained in a more organized and comprehensive manner.

4.      “HRT is limited by the reluctance that some women have about estrogens” -Some women may be hesitant to use HRT because of concerns about the safety of estrogens. Estrogen therapy has been linked to an increased risk of certain health conditions, such as breast cancer and blood clots. As a result, some women may be reluctant to use HRT, even if it could help alleviate their menopausal symptoms. This reluctance can limit the effectiveness of HRT as a treatment option. Please paraphrase the sentence to reflect the above.

Methods:

1.      It is necessary for the authors to provide justification for the exclusion of pelvic prolapse ≥ II grade in their study. Without this justification, it is not clear why this condition was excluded and how it may have affected the results of the study.

2.      It is important for the authors to explain why only diabetic patients were excluded from the study in terms of chronic inflammatory disease. Any other chronic conditions were excluded from the study?

3.      What do authors mean by “the pathologies of the urinary tract”?

4.      I would suggest that the authors include information about the ethnicity of the patients included in the study. This demographic information can provide valuable context and insight into the results of the study and can help to determine if certain ethnic groups may be more or less likely to experience certain symptoms or conditions.

5.      It is necessary for the authors to provide justification for the use of two-sample t-tests in their study. Without this justification, it is not clear why this statistical test was chosen and how it may have affected the results of the study.

Discussion

1.      Please correct AOB instead of OAB

2.      In terms of occurrence of the OAB, genetic factors should be highlighted too (https://www.ajog.org/article/S0002-9378(22)00616-0/fulltext)

3.      “In menopausal OAB patients, HRT has been demonstrated to significantly reduce symptom discomfort and is highly effective in alleviating genitourinary symptoms of menopause (GSM).” GSM acronym was already explained in the introduction part of the paper.

4.      Please include the following sentence in the discussion: Estrogen therapy is a treatment option for women who are experiencing symptoms of OAB. Estrogen works by improving the elasticity and strength of the muscles in the pelvic floor, which can help improve bladder control and reduce the frequency of urinary incontinence. In addition, estrogen therapy may help to decrease the symptoms of OAB by raising the bladder's sensory threshold. This means that estrogen can help to increase the amount of urine that the bladder can hold before the individual feels the need to urinate. By raising the bladder's sensory threshold, estrogen therapy may be able to reduce the frequency of urination and improve symptoms of OAB.

5.      Please add references for the following sentence: “Guidelines for the management of OAB do not currently include low-dose vaginal estrogen.”

Author Response

Reviewer 2: Response to Comments and Suggestions.

Introduction:                                                                                                                                    

Comment no. 1: " …. I find the introduction to be somewhat confusing and difficult to understand. It would be helpful if the authors rewrote the introduction in a more organized and coherent manner, explaining the main points and ideas they are trying to convey to the reader in a clear and concise way…".                                                                                                                                 Response to comment no. 1: We have re-organized the introduction. Hopefully the text is now relayed in a coherent  and concise manner.

Comment no. 2: The effectiveness of estrogen in treating urinary incontinence and overactive bladder symptoms has been studied in several clinical trials … estrogen therapy can be effective in improving symptoms of UI and OAB …by  improving the elasticity and strength of the muscles in the pelvic floor, which can help improve bladder control and reduce the frequency of urinary incontinence. …                                                                                                                                      Response to comment no. 2: conveyed in at the end of the introduction.

Comment no. 3: Menopause senescence … is a natural part of the aging process and is not considered a medical condition. Please elaborate that in the introduction.                                 Response to comment no. 3: Elaborated in the first paragraph of the introduction.

Comment no. 4: The term genitourinary syndrome of menopause (GSM) refers to a group of symptoms that can affect the genitals and urinary system in women who are going through or have gone through menopause. These symptoms can include dryness, itching, and burning in the vagina, as well as urinary incontinence and recurrent urinary tract infections. These all should be highlighted in the introduction section of the paper.                                                                  Response to comment no. 4: According to the above suggestion all relevant symptoms have been incorporated in the text.

Comment no. 5: I find the explanation of UI, GSM, and VVA in the introduction section of the paper to be unclear and lacking coherence…. It would be helpful if the terms were defined and explained in a more organized and comprehensive manner.                                                        Response to comment no. 5: UI, GSM and VVA have been defined as requested.

      Comment no. 6: “…Estrogen therapy has been linked to an increased risk of certain health conditions, such as breast cancer and blood clots. As a result, some women may be reluctant to use HRT, even if it could help alleviate their menopausal symptoms. … Please paraphrase the sentence to reflect the above.                                                                                                            Response to comment no. 6: Rephrased as required.

Methods:                                                                                                                                        

Comment no. 7:  It is necessary for the authors to provide justification for the exclusion of pelvic prolapse ≥ II grade in their study. Without this justification, it is not clear why this condition was excluded and how it may have affected the results of the study.                                                      Response to comment no. 7: Numerous studies have demonstrated overlap between POP and LUTS. Additionally, several trials of women with concomitant OAB and POP symptoms demonstrated symptom improvement following POP repair surgery, suggesting that POP may have a role in some women's overactive bladder (OAB) symptoms.By excluding POP >2 stage, we hope to “avoid dealing “ with OAB caused by significant prolapse that may play a differnt role in the pathofisiology of OAB.

Comment no. 8:  It is important for the authors to explain why only diabetic patients were excluded from the study in terms of chronic inflammatory disease. Any other chronic conditions were excluded from the study?                                                                                              Response to Comment no. 8:  We clarified that other chronic inflammatory disease were equally excluded.

Comment no. 9:  What do authors mean by “the pathologies of the urinary tract”?                        Response to Comment no. 9:  Omitted in the current text – does indeed seem odd.

Comment no. 10:  I would suggest that the authors include information about the ethnicity ofthe patients included in the study.

Response to Comment no. 10:  Patients were of Mixed ethnicity most of them Italian. Unfortunately, the number of patients is too small to deduce significant information related to ethnicity. 

Comment no. 11:  It is necessary for the authors to provide justification for the use of two-  sample t-tests in their study. Without this justification, it is not clear why this statistical test was chosen and how it may have affected the results of the study.                                                   Response to Comment no. 11:  We thouthfully considered the above and as such all stasistical analyses were re-performed using alternative tests as described in the methods section.

Discussion                                                                                                                                      Comment no. 12:  Please correct AOB instead of OAB.                                                         Response to comment no. 12: we did, Thank you

Comment no. 13:   In terms of occurrence of the OAB, genetic factors should be highlighted too (https://www.ajog.org/article/S0002-9378(22)00616-0/fulltext)                                                 Response to comment no. 13: The MS was cited and alluded to (reference no.30).

Comment no. 14: GSM acronym was already explained in the introduction part of the paper.                                                                                                                                      Response to comment no. 144: Thanks. Redundancy omitted.

Comment no. 15: Please include the following sentence in the discussion: Estrogen therapy is a treatment option for women who are experiencing symptoms of OAB. Estrogen works by improving the elasticity and strength of the muscles in the pelvic floor, which can help improve bladder control and reduce the frequency of urinary incontinence. In addition, estrogen therapy may help to decrease the symptoms of OAB by raising the bladder's sensory threshold. This means that estrogen can help to increase the amount of urine that the bladder can hold before the individual feels the need to urinate. By raising the bladder's sensory threshold, estrogen therapy may be able to reduce the frequency of urination and improve symptoms of OAB.

Response to comment no. 15: Thanks for the suggestion. Incorporated in the text as such: "Estrogen therapy is a treatment option for women who experience symptoms of OAB. Estrogen improves the elasticity and strength of the muscles in the pelvic floor thereby expanding bladder control and raises the bladder's sensory threshold thereby reducing urinary incontinence. Altogether, estrogen can help to reduce the frequency of urination and to manage OAB."

Comment no. 16: Please add references for the following sentence: “Guidelines for the management of OAB do not currently include low-dose vaginal estrogen.”                                   Response to comment no. 16: Added to the list of references no. 31.

Round 2

Reviewer 1 Report

The overall issues were amended by authors. Some minor errors or problems could be improved before acceptance.

1.          OAB is “overactive bladder”. (line 17)

2.          Authors should at least mention the results of reliability analysis of GPFSBQ that were conducted by recent studies related to postmenopausal OAB.

3.          The exclusion criteria of patients should be stated specifically, e.g., hypertension, chronic kidney disease.

4.          The characteristics of patients should include the data of urodynamic testing (table 1), or authors should at least explain why such information was not included.

Author Response

Reviewer 1 (Round 2)

The overall issues were amended by authors. Some minor errors or problems could be improved before acceptance.

  1. Comment no. 1: OAB is “overactive bladder” (line 17)

Response to comment no. 1: defined in line 14

  1. Comment no. 2: Authors should at least mention the results of reliability analysis of GPFSBQ that were conducted by recent studies related to postmenopausal OAB.

Response to comment no. 2: Provided as requested  - In a recent systematic review the reliability of GPFBQ was validated and graded "sufficient" for good measurement of properties (intraclass correlation coefficient greater than 0.7) – reference no. 20 introduced.

  1. Comment no. 3: The exclusion criteria of patients should be stated specifically, e.g., hypertension, chronic kidney disease.

Response to comment no. 3: added as requested.

  1. Comment no. 4: .The characteristics of patients should include the data of urodynamic testing (table 1), or authors should at least explain why such information was not included.

Response to comment no. 4: OAB is a clinical diagnosis and as such urodynamic testing is not deemed mandatory. A note was added in the methodology section.